# A linear reciprocal relationship between robustness and plasticity in homeostatic biological networks

**Tetsuhiro S. Hatakeyama**[1]*, **Kunihiko Kaneko**[1,2,3]

**1** Department of Basic Science, University of Tokyo, Meguro-ku, Tokyo, Japan, **2** Center for Complex Systems Biology, Universal Biology Institute, University of Tokyo, Komaba, Tokyo, Japan, **3** The Niels Bohr Institute, University of Copenhagen, Copenhagen, Denmark

* hatakeyama@complex.c.u-tokyo.ac.jp

**Data Availability Statement:** All relevant data are within the paper and its Supporting information files.

**Funding:** This work was partially supported by the Japan Society for the Promotion of Science (JSPS)

## Abstract

In physics of living systems, a search for relationships of a few macroscopic variables that emerge from many microscopic elements is a central issue. We evolved gene regulatory networks so that the expression of core genes (partial system) is insensitive to environmental changes. Then, we found the expression levels of the remaining genes autonomously increase to provide a plastic (sensitive) response. A feedforward structure from the non-core to core genes evolved autonomously. Negative proportionality was observed between the average changes in core and non-core genes, reflecting reciprocity between the macroscopic robustness of homeostatic genes and plasticity of regulator genes. The proportion coefficient between those genes is represented by their number ratio, as in the "lever principle", whereas the decrease in the ratio results in a transition from perfect to partial adaptation, in which only a portion of the core genes exhibits robustness against environmental changes. This reciprocity between robustness and plasticity was satisfied throughout the evolutionary course, imposing an evolutionary constraint. This result suggests a simple macroscopic law for the adaptation characteristic in evolved complex biological networks.

## Introduction

In recent decades, robustness in biological systems has been studied extensively in systems and quantitative biology [1–3]. Robustness refers to the maintenance of certain features or functions of biological systems against noise or changes in the environment [4–6]. Mechanisms for the robustness of specific gene expression also have been intensively studied [7]. In particular, housekeeping gene expression levels are believed to be robustly maintained across different environmental conditions [8]. Recent measurements, however, have shown that not all housekeeping genes are robust against environmental changes, but are only partially robust [9, 10]. Thus, investigations are needed to determine the possible limitation in the degree of robustness across genes, the constraints on global gene expression changes, and how these mechanisms have evolved [11–14].

(https://www.jsps.go.jp/english/) KAKENHI (17H06386, 20H00123) to K.K., by JSPS KAKENHI (21K15048) to T.S.H., by Novo Nordisk Fonden (https://novonordiskfonden.dk/) to K.K. The funders had no role in study design, data collection and analysis, decision to publish, or preparation of the manuscript.

**Competing interests:** The authors have declared that no competing interests exist.

Homeostasis refers to the constancy of "macroscopic" physiological quantities against environmental changes, such as body temperature and blood glucose level. Although the mechanism of homeostasis has often been attributed to interactions among few organs, many "microscopic" dynamics also play a role, including neurotransmission and gene expression. Here, we investigated the emergence of "macroscopic" homeostasis in a biological network consisting of multiple "microscopic" elements, inspired by recent studies demonstrating the relation between macroscopic thermodynamic quantities and microscopic molecular dynamics [15, 16]. Of course, the biological system is not in equilibrium, and microscopic elements therein are heterogeneous. Nevertheless, the evolved biological states are stable. By virtue of this stability similar to the equilibrium state in thermodynamics, some macroscopic laws between robustness and plasticity can be uncovered, with insensitivity of the homeostatic component and changeability of the remnant part in response to external changes. Indeed, previous studies demonstrated a linear relationship between robustness in the period and plasticity in the phase in the circadian rhythm [17, 18].

In this study, we explored gene expression dynamics governed by evolving a gene-regulatory-network structure numerically so that the expression level changes of core genes were insensitive to environmental changes. Then, a feedforward structure from the non-core part of the network (i.e., the regulator genes) evolved autonomously. The robustness in the expression of core genes and the plasticity in that of regulator genes showed a linear reciprocal relationship; the increase of the former was associated with the latter. The proportion coefficient between those genes is represented by their number ratio, as in the "lever principle", in which a decrease in the ratio results in a transition from perfect to partial adaptation, with only a portion of the genes exhibiting robustness. This result suggests a simple macroscopic law for the adaptation characteristic in evolved complex biological networks.

## Model and methods

Gene regulatory networks are among the most well-known examples of complex biological networks [11–13, 19–21], in which each gene activates or inhibits other genes, including self-regulation (Fig 1). In the present model, these interactions are represented as an interaction matrix, **J**; when the $j$th gene activates or inhibits the $i$th gene, $J_{ij}$ takes on a value of 1 or -1, respectively, and when there is no interaction, $J_{ij}$ is 0. Inputs from the environment were further introduced to study adaptation against environmental changes, which globally regulate

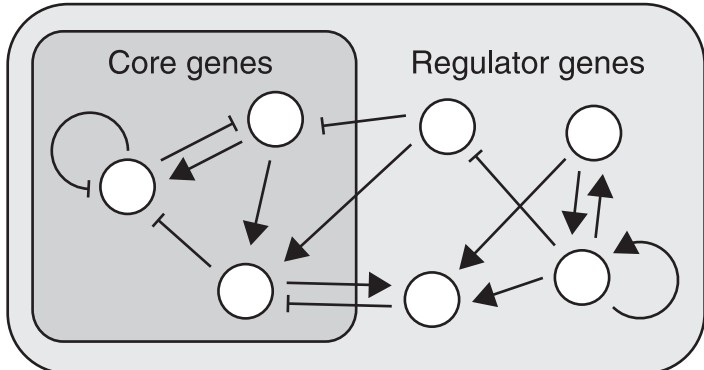

**Fig 1. Schematic representation of the gene regulatory network.** Each white circle represents a gene. Genes regulate the expression of other genes (including self-regulation). Triangular and flat arrowheads represent activating and inhibitory interactions, respectively.

the expression of all genes. The environmental inputs are multi-dimensional, and each gene can exhibit a different degree of sensitivity to these inputs in two directions, represented as $h_i$ = 1 or −1 for simplicity. Sign of $h_i$ is randomly chosen and fixed during each evolutionary process. The environmental changes are represented by a single parameter, $\alpha$. If signs of $\alpha$ and $h_i$ are identical (different), the $i$th gene is activated (inhibited). We consider on/off-type gene expression dynamics with a given threshold: if the total input exceeds the threshold value, $y_{\text{th}}$, the genes are turned on. Further, the expression level of each gene can also fluctuate due to noise. These dynamics are given by the following stochastic differential equations:

$$\frac{dx_i}{dt} = \frac{1}{1 + \exp(-\beta(y_i - y_{\text{th}}))} - x_i + \epsilon + \eta_i(t),$$

$$y_i = \sum_j \frac{J_{ij} x_j}{\sqrt{N}} + \alpha h_i, \tag{1}$$

$$< \eta(t) >= 0, < \eta_i(t)\eta_j(t') >= \sigma^2 \delta(t - t')\delta_{ij},$$

where $x_i$ is the expression level of the $i$th gene, $\beta$ is the steepness of gene induction around the threshold (i.e., when $\beta$ is sufficiently large, the gene expression dynamics approach on/off-type switching, and $\beta$ corresponds to the Hill coefficient in a model that is often adapted in biology), and $N$ is the total number of genes; the interaction term is scaled by $\sqrt{N}$ considering the scale in random variables. $\epsilon$ is a small spontaneous induction, whose value does not change the result as long as it is much smaller than 1, and $\eta_i(t)$ is the Gaussian white noise in the gene expression level. Of note, our model is quite similar to a neural network [20], and we can easily extend our results to other complex biological networks. Here, we set $y_{\text{th}}$ to 0.3, $\beta$ to 20.0, $\epsilon$ to 0.05, and $\sigma^2$ to 0.01.

We first investigated the adaptation dynamics involving a large number of components. In general, when the environment changes, organisms do not maintain all of the components constant but rather need to sustain only a portion of these essential components. Indeed, in adaptation experiments, the expression of only a portion of the genes in the network could be robustly maintained against environmental change, whereas the expression levels of most other genes were altered [22]. This is natural, because maintaining an entire system completely unchanged is impossible when each element is sensitive to the environment. To investigate the characteristics of these adaptation dynamics, we considered the following simple situation: some components behave as a core of homeostasis, while others function as regulators to maintain the core robustly against environmental changes. Accordingly, we designate genes incorporated in the core as core genes and the others as regulator genes (see Fig 1).

We then optimized the network structure **J** to achieve robustness of the homeostatic core by mimicking the evolutionary process [11, 12, 14, 23, 24]. From mutants with a slight change in **J**, we selected those exhibiting higher robustness in the expression of core genes to environmental changes for the next generation, as parameterized by $\alpha$. First, the condition without an environmental stimulus was represented by $\alpha = 0$. The system was then allowed to relax to a steady state to obtain the expression pattern $\{x_i^{\text{st}}(0)\}$, where $x_i^{\text{st}}(\alpha')$ is a steady-state value of $x_i$ at $\alpha = \alpha'$. Steady-state value $x_i^{\text{st}}$ was calculated by averaging $x_i$ over 1,000 time steps after the relaxation for each genotype at each generation. We then changed $\alpha$ to both positive and negative values ($\alpha_1$ and $-\alpha_1$) and let the system in each case relax to the steady-state again to obtain $\{x_i^{\text{st}}(\pm\alpha_1)\}$. Here, we set $\alpha_1$ to 1.0. To analyze the robustness and plasticity of the gene expression levels, we calculated the average change in the expression level of genes in different

environments as follows:

$$\Delta X^{\mathrm{C}}(N^{\mathrm{C}}) = \sum_{i \in \mathrm{core}} \sum_{\alpha \in \{\alpha_1, -\alpha_1\}} \frac{\left(x_i^{\mathrm{st}}(\alpha) - x_i^{\mathrm{st}}(0)\right)^2}{2N^{\mathrm{C}}}, \tag{2}$$

$$\Delta X^{\mathrm{R}}(N^{\mathrm{R}}) = \sum_{i \in \mathrm{regulator}} \sum_{\alpha \in \{\alpha_1, -\alpha_1\}} \frac{\left(x_i^{\mathrm{st}}(\alpha) - x_i^{\mathrm{st}}(0)\right)^2}{2N^{\mathrm{R}}}. \tag{3}$$

An individual with a smaller $\Delta X^{\mathrm{C}}$ shows higher robustness in the core and is assumed to have higher fitness. Reproduction was asexual. Then, the $k$th individual with $\Delta X_k^{\mathrm{C}}$ can produce an offspring with probability $P_k$, given as

$$P_k = \frac{\exp(-\beta_{\mathrm{evo}} \Delta X_k^{\mathrm{C}})}{\sum_l \exp(-\beta_{\mathrm{evo}} \Delta X_l^{\mathrm{C}})}, \tag{4}$$

where $\beta_{\mathrm{evo}}$ is the strength of the selection pressure. The networks **J** at the 0th generation are chosen randomly as described below. In each generation, each element in the offspring's **J** is changed among $\{\pm 1, 0\}$ with probability $p_{\mathrm{mut}}$. We set $\beta_{\mathrm{evo}}$ and $p_{\mathrm{mut}}$ to 40.0 and 0.01, respectively.

Note that here the steady state without an environmental stimulus did not change much by evolution, so that the evolution of insensitivity against environmental changes to be observed was not due to the change in the steady state. This is because we set the standard deviation of $\eta$ and $\epsilon$ much smaller than the threshold. When $\alpha$ is zero, all genes are activated only by the noise given by $\eta$ and spontaneous induction given by $\epsilon$. Then, all genes are normally turned off and fluctuate around $x_i = \epsilon$ almost independently of **J**; since $\epsilon/\sqrt{N}$ is 0.005, inputs from 60 genes (60% of the total genes) would have to concentrate on a given gene to exceed the threshold and to turn on a gene constantly without $\alpha$. Such concentration hardly occurs due to the sparseness of networks. Hence, the average gene expression without input did not change through an evolutionary process, and all genes were kept to be off initially. In addition, since the threshold value was fixed, the sensitivity of genes against environmental changes was kept constant. Indeed, in Fig 2A (before evolution) and Fig 2B (after evolution), gene expression levels fluctuated around $\epsilon$ without input (time = 0–100), and those of the core transiently increased in response to changes in $\alpha$ (at time = 100 and 200), indicating that those genes were kept to be sensitive against environmental changes. This setting might seem biologically unrealistic, but we here focus on the evolution of responses against environmental changes and not on the nature of the steady-state gene expression pattern unlike previous studies (for example [25]). If we adopt constant selection for an arbitrary expression pattern, it will be difficult to distinguish the effect of the evolution of responses from that of the steady state. Thus, we adopt the present simplest settings, in which gene expression pattern is quite stable throughout evolution, and only responses can evolve.

In this study, we set $N$ to 100 and the total number of individuals $M$ to 300, unless otherwise noted. Initially, the elements in **J** take a value of 1 or -1 with a probability of $p_{\mathrm{link}}$ set to 0.1, and take 0 with a probability of $p_{\mathrm{link}}$. We changed the fraction of the core genes to the whole genes, $N^{\mathrm{C}}/N$, from 0.05 to 1.0 and investigated the dependence of the behavior of evolved gene expression dynamics on $N^{\mathrm{C}}/N$. We repeated the simulation 100 times for each parameter set and plotted the average of those.

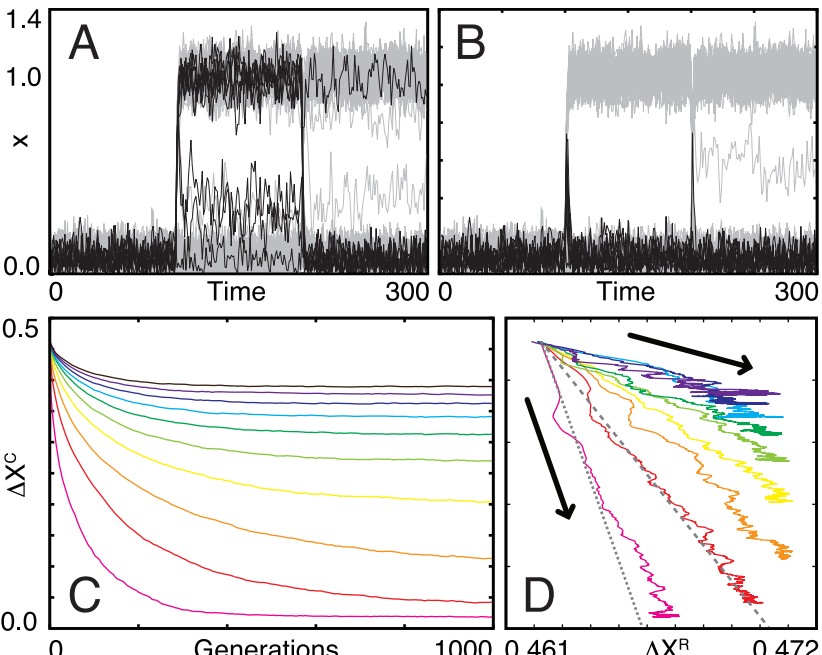

**Fig 2. Evolutionary process of gene regulatory networks.** (A, B) Adaptation dynamics of genes ($x_i(t)$) of an individual with the highest fitness before (A: 0th generation) and after (B: 1000th generation) evolution. $\alpha$ was changed from 0 to 1 and from 1 to -1 at time 100 and 200, respectively. Black and gray lines indicate the time course of the core ($N^C = 10$) and regulator ($N^R = 90$) genes, respectively. (C) Changes in $\Delta X^C$ from the 0th to 1000th generations and (D) the corresponding trajectory at the $\Delta X^R$–$\Delta X^C$ plane. All of the trajectories start from the same point ($\Delta X^C = \Delta X^R = \Delta X_0 \simeq 0.462$). Different color lines indicate evolutionary trajectories with different $N^C/N$: magenta for 0.1, red for 0.2, orange for 0.3, yellow for 0.4, lime for 0.5, green for 0.6, cyan for 0.7, blue for 0.8, purple for 0.9, and brown for 1.0. Gray dotted and dashed lines are given by Eq 5 for $N^C = 10$ 10 and 20, respectively.

## Results and discussion

For all $N^C$ values, the network structures evolved to decrease $\Delta X^C$ (see Fig 2A and 2B as an example), whereas the final evolved state depended on $N^C$ (Fig 2C). When $N^C$ was sufficiently small, $\Delta X^C$ reached a steady value in the early generations, which was close to zero; that is, the core genes showed perfect adaptation [26, 27]. In contrast, when $N^C$ was large, its steady-state value was larger than zero; that is, adaptation was only partial. This value increased with $N^C$. The system showed a transition from perfect to partial adaptation at $N^C = N^{C*}$, which lies between 20 and 25. It is noteworthy that the relaxation was slowed down for $N^C/N = 0.2$ to 0.6, in particular between $N^C/N = 0.2$ and 0.3. In statistical physics, it is commonly observed that the relaxation of systems to the equilibrium is slowed down in the vicinity of the transition point. Then, the observed slowing down will be the sign of critical slowing down and provide evidence of a transition between perfect and partial adaptation regimes. Note that even when $N^C$ was equal to $N$ (i.e., without the regulatory genes), $\Delta X^C$ still decreased slightly throughout evolution; that is, the networks can show intrinsic robustness without regulators. We define this $\Delta X^C$ value for the case of $N^C = N$ as $\Delta X_{int}$.

Interestingly, as $\Delta X^C$ decreased during evolution, $\Delta X^R$ increased almost monotonically in all cases (see Fig 2D). This result implies that under evolutionary selection, to increase the robustness of the expression of core genes, the plasticity in the expression of regulatory genes simultaneously increases. Here, the evolutionary trajectories in the space of $\Delta X^C$ and $\Delta X^R$ showed nearly linear behavior (Fig 2D). Evolution then stopped either when $\Delta X^C$ reached

approximately zero or when $\Delta X^{\mathrm{R}}$ increased and reached a certain threshold value, $\Delta X^{\mathrm{R}^*}$ ($\simeq$ 0.471). Of note this threshold value depends on the stochasticity in gene expression dynamics. If there is no fluctuation, the steady state value of each $x_i$ is $\epsilon$ without environmental input, and $1 + \epsilon$ under the input $\alpha$ with the same sign to $h_i$. Because half of the genes have positive $h_i$, and another half have a negative one, the threshold value would be estimated as 0.5 (= $(1 + \epsilon - \epsilon)^2$/ 2.0). However, because of fluctuation and a constraint to keep gene expression positive, the steady state value of $x_i$ without input will be higher than $\epsilon$; $x_i$ can reach about 0.3 by fluctuation but is truncated at 0 at the bottom, as shown in Fig 2A and 2B, where we set $\epsilon$ as 0.05. Then, the noise in gene expression is asymmetric in the vicinity of $x_i = 0$. Thus, the threshold value cannot reach 0.5 and will be slightly lower than 0.5. This suggests that there is an upper limit at which the regulator can buffer the changes in the core. If the buffering capacity is reached through evolution, the compensation by the regulators is not sufficient to allow for perfect adaptation of the core.

Here, values of $\Delta X^{\mathrm{C}}$ (evolved from 0.462 to 0 at most) and $\Delta X^{\mathrm{R}}$ (evolved from 0.462 to 0.473) were asymmetrical, which was due to the setting to make gene expression be zero without input and to be one only under sufficiently strong input. Since $\alpha_1$ was set to be larger than the threshold $y_{\mathrm{th}}$, half of the genes whose $h_1$ had the same sign as $\alpha$ were strongly activated. Then, at the beginning of evolution, when each gene regulatory network was random, only some, but not many, genes were suppressed if inhibitory regulations were concentrated on them. We set $\alpha_1$ and $y_{\mathrm{th}}$ to 1 and 0.3, respectively, and then if seven inhibitory interactions were concentrated, core genes would be suppressed. Those genes' expression levels would be intermediate between 0 and 1. Indeed, in Fig 2A, before evolution, some genes showed intermediate expression level, whereas those of almost all activated genes were around 1. Thus, when the environmental input was sufficiently strong, there was an asymmetry between the numbers of fully activated genes (whose expression level is almost 1) and intermediately activated genes (whose expression level is less than 1). When systems evolved to make robust core genes, inhibition of a large number of fully activated genes mainly contributed to $\Delta X^{\mathrm{C}}$. In contrast, activation of a small number of intermediately activated genes mainly contributed to $\Delta X^{\mathrm{R}}$. Therefore, the asymmetry between $\Delta X^{\mathrm{C}}$ and $\Delta X^{\mathrm{R}}$ emerged.

Evolved networks have distinct structures. The number of inhibitory interactions to the core genes from both the core and regulator increased (Fig 3A and 3B). In particular, when $N^{\mathrm{C}}$ was small, inhibitory interactions from the regulators prominently increased, whereas those from the core increased only slightly. By contrast, when $N^{\mathrm{C}}$ was large, the number of inhibitory interactions from the core also increased significantly. This suggests that regulation from the regulator is a primary driving force of homeostasis for small $N^{\mathrm{C}}$, whereas for large $N^{\mathrm{C}}$ (i.e., small $N^{\mathrm{R}}$), the core itself also functions in maintaining homeostasis. Indeed, even when all of the interactions from the regulators were removed from the evolved networks, $\Delta X^{\mathrm{C}}$ still decreased when $N^{\mathrm{C}}$ was large (Fig 3E). $\Delta X^{\mathrm{core}}$ started to decrease at around $N^{\mathrm{C}} \simeq 25–30$. In the perfect adaptation region, the core genes were perfectly suppressed by regulatory genes, whereas if the regulation from the regulatory genes was lacking, the core genes never showed adaptation (below $N^{\mathrm{C}} \simeq 30$). In contrast, in the partial adaptation region, the core genes regulated each other because the number of regulatory genes was insufficient. Then, these showed partial robustness without the regulation from the regulatory genes (above $N^{\mathrm{C}} \simeq 30$). In contrast, the number of interactions from the core or regulator to the regulator changed only slightly compared with that of the initial random network (Fig 3C and 3D).

In the evolved networks, the propagation of perturbation also showed distinct changes from the random network. We analyzed how local perturbation to a gene propagates to the entire network; we changed the sign of $h_i$ for a single gene and then counted the number of genes that were flipped between the on and off states. As shown in the cyan and red circles in

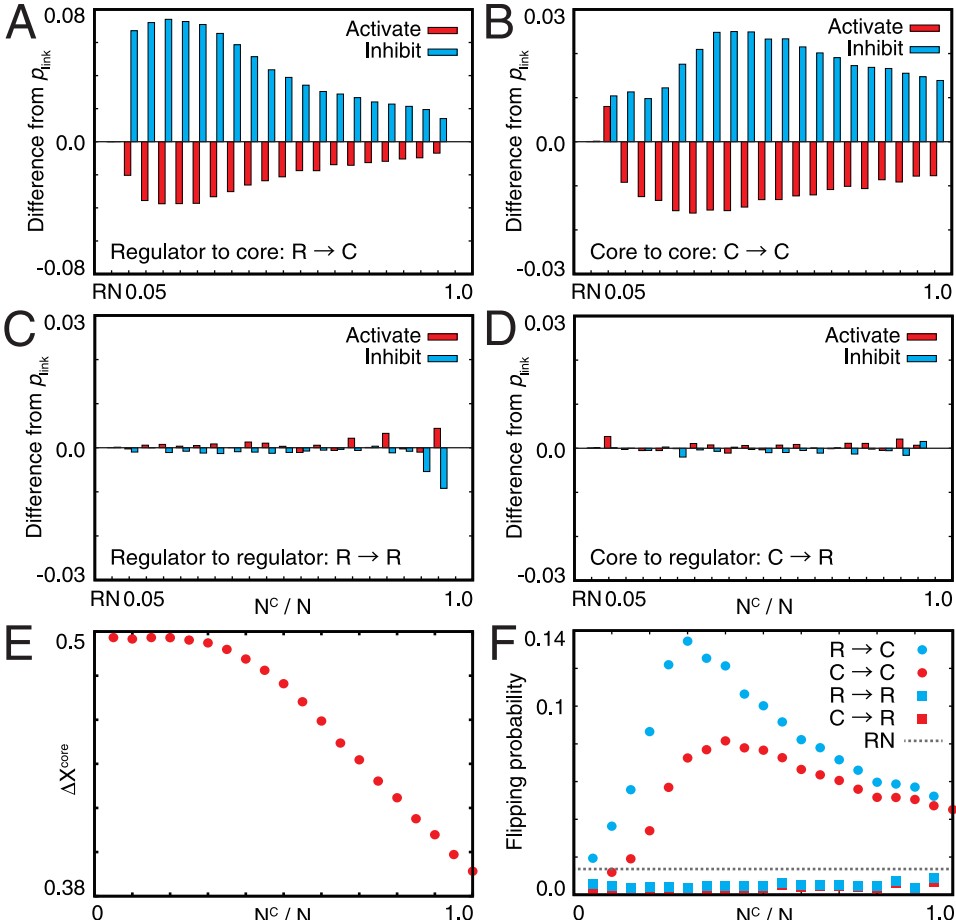

**Fig 3. Interactions between the core and regulator genes in evolved networks with varied $N^C$.** (A–D) Difference of the linking probabilities between two nodes in the evolved networks from the default value $p_{\text{link}}$. RN indicates the random network. Each graph shows the linking probabilities (A) from the regulator to the core, (B) from the core to the core, (C) from the regulator to the regulator, and (D) from the core to the regulator. Red and cyan bars represent the linking probability for activating and inhibitory interactions, respectively. (E) $\Delta X$ of the core without every interaction from the regulator. (F) Flipping probabilities of each node from the off to on state or from the on to off state after a change in the sign of $h_i$. Each flipping probability is averaged for every node. Cyan circles and squares represent the flipping probabilities of nodes in the core and the regulator for a change in a node in the regulator, respectively. Red circles and squares represent these flipping probabilities for a change in a node in the core, respectively. The gray dotted line represents the flipping probability measured for the random network.

Fig 3F, the flipping probability of each gene in the core increased with the increase in $N^C$, which reached the maximal level at around $N^C \simeq 30$. This will be related to the transition. If $N^C$ increased, the contribution of each regulatory gene to repress genes in the core needed to be increased, and then perturbation of one regulatory gene had a great influence. However, if $N^C$ was further increased above the transition point, the mutual regulation within the core genes turned to be relatively stronger than the regulation from the regulatory genes, then the influence of a flip of one regulatory gene decreased. Thus, we expect that the influence of perturbation will show the peak around the transition point from perfect to partial adaptation.

Note that the number of links to the regulator did not change (Fig 3C and 3D). Nevertheless, the flipping probability of each gene in the regulator decreased throughout evolution (see the cyan and red squares in Fig 3F). This result indicates that the expression of each regulator

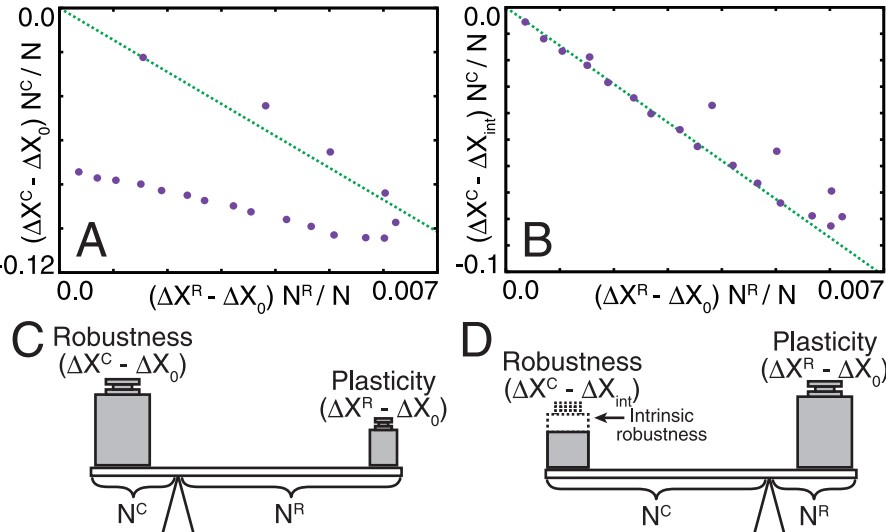

**Fig 4. Relationships between robustness and plasticity.** (A, B) Total change in gene expression in the core plotted against that in the regulator. Averaged values of $\Delta X^C$ and $\Delta X^R$ through 100 generations from the 900th to 1000th generation are used as the steady-state value. The difference of $\Delta X^C$ from $\Delta X_0$ and from $\Delta X_{int}$ is plotted in (A) and (B), respectively. Green dotted lines are drawn to fit the points for $N^C > 30$ in (B). Two lines pass through the origin and have the same slope for both (A) and (B). (C, D) Lever principle for the robustness-plasticity relationship.

gene behaves more independently than those in the random network, which could increase the plasticity in the regulator genes, as shown in Fig 2D.

Then, we analyzed the quantitative relationship between $\Delta X^C$ and $\Delta X^R$ in the evolutionary steady state. The total change in gene expression in the core, $N^C(\Delta X^C - \Delta X_0)$, and in the regulator, $N^R(\Delta X^R - \Delta X_0)$, where $\Delta X_0$ is $\Delta X$ of the random network, is plotted in Fig 4A. For $N^C < N^{C*}$, where perfect adaptation occurs, the following linear relationship was found:

$$N^C(\Delta X^C - \Delta X_0) \simeq -aN^R(\Delta X^R - \Delta X_0) \tag{5}$$

with $\Delta X^C \simeq 0$ due to perfect adaptation, where $a$ is a positive constant ($a \simeq 14.3$). In contrast, for large $N^C$, only partial adaptation could be achieved, and $\Delta X^C$ remained finite but was still smaller than $\Delta X_{int}$, the value for $N^R = 0$ (i.e., the case without the regulator). The difference $\Delta X^C - \Delta X_{int}$ ($< 0$) for $N^C$ is supported by the plasticity of the regulator; that is, the increment of $\Delta X^R$ from the random case. Indeed, for large $N^C$, we found the following linear relationship (see Fig 4B):

$$N^C(\Delta X^C - \Delta X_{int}) \simeq -aN^R(\Delta X^R - \Delta X_0). \tag{6}$$

Again, to achieve the decrease in $\Delta X^C$, $\Delta X^R$ changes more following the linear rule, where $a$ takes on the same value as shown in Eq 5.

Interestingly, the linear relationship in Eq 5 was maintained throughout the evolutionary course. The time course of $(\Delta X^C, \Delta X^R)$ satisfied Eq 5 as long as $N^C < N^{C*}$ (see gray dotted and dashed lines in Fig 2D). This indicates that the linear relationship imposed a constraint at any evolutionary time point. Note that, in the case of large $N^C$, the evolutionary trajectory did not follow the linear relationship because the intrinsic robustness evolved; in Eq 6, $\Delta X^C$ approaches $\Delta X_{int}$ as $N^R \to 0$, whereas the evolutionary trajectory started from $\Delta X_0$. If the evolution of the regulations from the regulatory genes were much faster than the intrinsic robustness, the linear

relationship would hold in the early stage. However, the timescale of evolution depends on $N^C$; if $N^C$ is much larger than $N^R$, the number of intra-connections among the core genes is much larger than that of interconnections between core and regulatory genes and they can change with higher probability by the mutation in each generation. Then, intra-connections will typically change faster than the interconnections and evolve faster. This will be why we cannot observe the clear linear relationship for large $N^C$.

Therefore, we demonstrated linear relationships between robustness in the homeostatic core and plasticity in the regulator in evolved networks. Specifically, when the system shows higher robustness, it shows higher plasticity. If the fraction of the core is large, the homeostatic core will achieve only partial (i.e., not perfect) adaptation. Nevertheless, the linear relationship holds with the same proportion coefficient as in the perfect adaptation case.

Although the derivation of the linear relationships (Eqs 5 and 6) requires further study, an analogy with the lever principle may provide a more intuitive interpretation. For $N^C < N^{C^*}$, all genes in the core show perfect adaptation and $\Delta X^C$ approaches $\sim 0$, for which the total plasticity in regulator genes $N^R(\Delta X^R - \Delta X_0)$ compensates for the original change in the core $N^C \Delta X_0$ (Fig 4C). Then, if the number of plastic genes required to reach a balance exceeds $N^R$, the plasticity of the regulator genes is not sufficient to cancel out changes in the core, and adaptation is only partial. In the latter scenario, the intrinsic robustness conferred by regulation from the core evolved (Fig 3E), so that "the weight" for compensation is deducted, whereas the action by the regulator is maintained (Fig 4D). Therefore, the linear relationship with the same coefficient also holds for the case of partial adaptation.

The coefficient $a$ in the lever rule is estimated by noting that at the transition point from perfect to partial adaptation ($N^C = N^{C^*}$, $N^R = N^{R^*}$), the regulator genes are fully plastic, whereas $\Delta X^C = 0$ is maintained. Then,

$$a = -\frac{N^{C*}}{N^{R*}}\frac{(0 - \Delta X_0)}{(\Delta X^{R*} - \Delta X_0)}$$

Noting that $N^{C^*} \simeq 22$, $N^{R^*} \simeq 78$ according to Fig 2, and recalling $\Delta X^{R^*} \simeq 0.471$ and $\Delta X_0 \simeq 0.462$, $a$ is estimated as $a \simeq 14.3$, which agrees well with the observed value.

Interestingly, the lever rule (Eq 5) is also valid for the evolutionary process. Consistency between evolutionary trajectories and the dependence of $N^C$ on the stationary state in Eq 5 indicates that evolutionary progress satisfies the balance between robustness and plasticity. Hence, the same macroscopic law governs both evolutionarily optimized states and their evolutionary trajectories.

Finally, we investigated the system-size dependence of the robustness by changing the total number of genes and numerically evolved the system (Fig 5). We found that the robustness of the core, $\Delta X^C - \Delta X^0$, was scaled by $(N^C/N)N^{3/4}$; if only the ratio of $N^C$ to $N$ were the important control parameter, robustness should be scaled by $(N^C/N)N^0$, or if only the absolute number of the core genes were important, it should be scaled by $(N^C/N)N^1$. The actual scaling found numerically is intermediate between these two cases; the maximum core size that shows perfect adaptation increases with the total number of genes, but not in proportion to it, but with the power 3/4. This suggests that the balance between core and regulatory genes depends not simply on the number of genes.

The lever principle may impose a fundamental constraint on homeostasis. Previous analyses of gene expression changes in response to environmental stress revealed that the expression levels of some genes change transiently and then return to the original level, whereas those of others change continuously [22], corresponding to the core and regulator of our model, respectively. Interestingly, experimental data suggest that the total change in gene expression

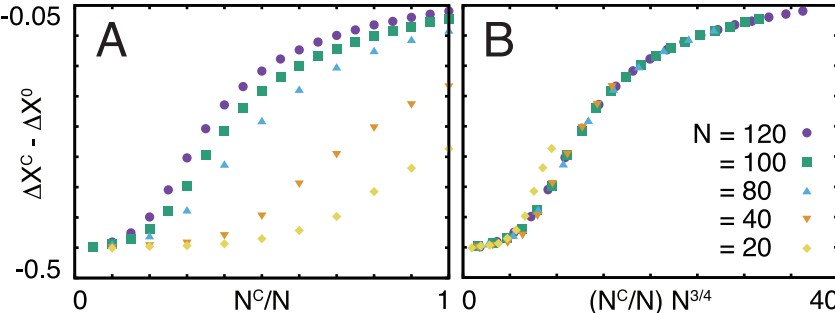

**Fig 5. System-size dependence of the robustness.** Dependence of $\Delta X^C - \Delta X^0$ on (A) $N^C/N$ and (B) $(N^C/N)N^{3/4}$ for systems with various numbers of genes.

in the steady state is proportional or correlated to its transient change, which is similar to the reciprocity between robustness and plasticity according to the lever rule uncovered with our model. Further studies will be required to reveal how the lever rule emerges and if the rule can be generalized to other homeostatic behaviors in biology.

## Supporting information

**S1 Data.**
(ZIP)

**S2 Data.**
(ZIP)

## Author Contributions

**Conceptualization:** Tetsuhiro S. Hatakeyama, Kunihiko Kaneko.

**Formal analysis:** Tetsuhiro S. Hatakeyama.

**Funding acquisition:** Tetsuhiro S. Hatakeyama, Kunihiko Kaneko.

**Investigation:** Tetsuhiro S. Hatakeyama.

**Methodology:** Tetsuhiro S. Hatakeyama.

**Project administration:** Tetsuhiro S. Hatakeyama.

**Software:** Tetsuhiro S. Hatakeyama.

**Supervision:** Tetsuhiro S. Hatakeyama, Kunihiko Kaneko.

**Validation:** Tetsuhiro S. Hatakeyama, Kunihiko Kaneko.

**Visualization:** Tetsuhiro S. Hatakeyama.

**Writing – original draft:** Tetsuhiro S. Hatakeyama.

**Writing – review & editing:** Tetsuhiro S. Hatakeyama, Kunihiko Kaneko.

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
