## [Decision Letter · Decision Letter 0]

14 Jul 2022

PONE-D-22-14450A Linear Reciprocal Relationship Between Robustness and Plasticity in Homeostatic Biological NetworksPLOS ONE

Dear Dr. Hatakeyama,

Thank you for submitting your manuscript to PLOS ONE. After careful consideration, we feel that it has merit but does not fully meet PLOS ONE’s publication criteria as it currently stands. Therefore, we invite you to submit a revised version of the manuscript that addresses the points raised during the review process.

First, I want to apologize for the somewhat slow review process. It was more difficult than usual to find the required number of reviewers. Second, when preparing your revision, I want to encourage you to take a close look at the PLOS ONE publication criteria, https://journals.plos.org/plosone/s/criteria-for-publication. Points 3-5 have all been found to be insufficient by the reviewers.

We look forward to receiving your revised manuscript.

Kind regards,

Claus Kadelka

Academic Editor

PLOS ONE

Journal Requirements:

[This research was partially supported by a Grant-in-Aid for Scientific Research (A) 431  (20H00123) and Grant-in-Aid for Scientific Research on Innovative Areas (17H06386) from the Ministry of Education, Culture, Sports, Science and Technology (MEXT) of Japan]

 [This research was partially supported by a Grant-in-Aid for Scientific Research (A) 431 (20H00123) and Grant-in-Aid for Scientific Research on Innovative Areas (17H06386) from the Ministry of Education, Culture, Sports, Science and Technology (MEXT) of Japan. The funders had no role in study design, data collection and analysis, decision to publish, or preparation of the manuscript.]

Reviewers' comments:

Reviewer's Responses to Questions

**Comments to the Author**

1. Is the manuscript technically sound, and do the data support the conclusions?

Reviewer #1: Yes

Reviewer #2: Partly

2. Has the statistical analysis been performed appropriately and rigorously? 

Reviewer #1: Yes

Reviewer #2: N/A

3. Have the authors made all data underlying the findings in their manuscript fully available?

Reviewer #1: No

Reviewer #2: No

4. Is the manuscript presented in an intelligible fashion and written in standard English?

Reviewer #1: Yes

Reviewer #2: No

5. Review Comments to the Author

Reviewer #1: Review of "A linear reciprocal relationship between robustness and plasticity in homeostatic biological networks" by T. Hatakeyama and K. Kaneko, PLoS One D-22-14450.

The manuscript is a short report about the evolution of gene expression sensitivity to environmental disturbances in a theoretical gene network model. The authors simulated the evolution of a gene network when a set of genes, named "Core genes", were selected to be expressed at a constant rate whatever the environment. Another set of genes, the "Regulator genes", could evolve in order to help core genes to become stable. The authors shows that core genes were indeed able to evolve robustness, but this evolution was associated with a proportional increase in the sensitivity of regulator genes to the environment. The explanation proposed in the manuscript is that there is a global robustness-plasticity balance in the network, which can be distributed among core and regulatory genes.

As far as I can tell, the authors' results are original, the manuscript is clearly and carefully written (see below for a list of possible misunderstandings), and the conclusions are well-supported. On the down side, the paper is a short report, relatively technical, with little background, and virtually no discussion; it is thus up to the reader to replace the paper in the field, which limits the readership to a relatively small community, and makes it impossible for a non-specialist to assess the interest of the authors' results. As I recently happened to be interested in the very same question in similar theoretical gene network models, I found the results both interesting and insightful, but I am afraid that this opinion might be a bit confidential.

I have a major concern, and some detailed comments.

**** Major concern ****

1) The reference expression x^st(0) for the fitness function (2) conditions the optimal gene expression of the network, but I could not understand how it was determined. Was it computed from an arbitrary genotype at the beginning of the simulation, or was it recomputed for all genotypes? If the first option is true, then the authors need to make it much more explicit. The second option has some deep implications that need to be considered thoroughly. If selection only acts on the robustness of gene expression irrespective to the gene expression level, all non-plastic genotypes are equivalent. However, this will drive the evolution of gene expressions; because of the sigmoid scaling of the gene expression regulation (equation (1)), the sensitivity of the network to the environmental input depends directly on the distance between the sum of J_ij x_j and the threshold. If there is no selection on gene expression, evolution will proceed by maximizing or minimizing (probably minimizing) J_ij x_j to put it as far as possible from the threshold. As a consequence, a network where all core genes were off could be very robust. This could be an explanation for the pattern in fig 3A and 3B, as inhibiting interactions to core genes seem to largely dominate. Yet, I am not sure that a network that would stabilize its expression by shutting down all core genes would be biologically realistic. I assume the authors could track the evolution of average gene expression to ensure that the network is not progressively shutting down to ensure robustness. Constant selection for an arbitrary expression pattern (i.e., drawing randomly the optimal x for each gene at the beginning of the simulation) is also an option. Citing and discussing Siegal & Bergman 2002 (PNAS 99(16), 10528-10532) is probably relevant here.

**** Details ****

2) I am not sure that enough details are provided to reproduce the simulations. Some details of the numerical procedure were implicit, and it is probably better if the authors made them explicit. For instance, were numerical simulations repeated several times? They might be affected by a substantial amount of genetic drift, as the population size is rather small (M=300). How was the "steady state expression" obtained? Since the differential equations were stochastic (sigma^2 > 0 in equation (1)), expression never reach a perfect equilibrium (there is always a chance to switch genes on and off randomly), and some threshold was probably used. Such a steady state might not exist for large values of sigma^2, as the network may regularly switch between different equilibria when disturbed. Was reproduction clonal? Was the steady state recomputed in non-mutated offspring? (because of the expression stochasticity, clones might have a slightly different fitness).

As a side note, the authors indicated that "All data are within the manuscript and the Supporting Information files"; I might have missed them but I could not find the corresponding links (to the data and/or the computer code).

3) The first part of the results (lines 66 to 84) probably belong to the methods.

4) Fig 2A and 2B: what does the y axis (labelled "x") stand for? Is it x(i) (gene expression?) or DeltaX (as suggested in the text line 90?).

5) I was expecting that the authors discuss the fact that DeltaXc evolves a lot (-100% to -10% depending on Nc) while DeltaXr evolves on a very small scale (0.461 to 0.471, which is about 2%). If plotted on the same scale, the evolution of DeltaXr would be invisible in panel 2D. This asymmetry is not well-represented in fig 4; the drawing features similar weight differences for R genes vs C, while in reality it would be impossible to distinguish the weight difference for regulatory genes.

6) Line 161, the authors develop an interesting argument: the critical number of core genes is reached when there is not enough R genes to balance the evolution of plasticity. The idea that the ratio Nc/Nr rather than the absolute number of core genes Nc drives the evolution of plasticity is an interesting idea, but this idea could be directly checked by running simulations with a higher number of R genes while maintaining Nc constant. Alternatives include explanations based on genetic load and/or genetic drift (e.g. natural selection in small populations is not strong enough to force robustness on more than Nc genes). If this was true, then Nc should depend on M, not on Nr.

Signed: Arnaud Le Rouzic

Reviewer #2: Reviewer's report

Summary:

The authors conducted evolutional simulations for a model of gene regulatory networks, which consisted of the core genes and the regulatory genes. As a fitness, they required the homeostasis of the core genes. For the evolved networks, they found that the robustness (homeostasis) of the core genes and plasticity of the regulatory genes were reciprocally correlated. Thus, the robustness of the core genes is compensated by the plasticity of the regulatory genes. In case the ratio of core genes is higher than the threshold value, this compensation was incomplete. They also observed the lever principle held between these two.

I think these results are novel and interesting enough, and will induce future research both in computational and experimental biology. Thus, it is worth publishing in PLOS ONE, provided the following points are made clear.

Major

L.96

The authors write that there is a transition from the perfect adaptation to the partial adaptation, and the critical value of Nc lies between 20 and 25. However, I do not observe the transition in Fig.2C; the lines for Nc/N=0.2 to 0.6 do not reach steady-state within 1000 generations and thus it is difficult to distinguish whether the steady-state value gradually changes or there indeed is the transition. This point should be clarified, for example by presenting the steady-state values for Nc/N=0.2 to 0.6. Whether this transition point agrees with Nc/N that Delta_X_R reaches the threshold value before Delta_X_C reaches zero should be clarified (although written briefly in L.106).

L.106

The authors write that there is a threshold value near 0.471 and I notice there indeed is the threshold. Intuitively, this threshold value is a consequence of the stochasticity of evolution and depends on beta_evo. Do you think this intuition correct, or there is other reason for this threshold? The author should mention the meaning or origin of this threshold value at least qualitatively.

L.127

In Fig.3E, the value starts to decrease near Nc=30, and in Fig.3E, peaks are also near Nc=30. Is this a mere coincidence or do they have some common reason? Also, Nc=30 is close to the transition point mentioned before. Is this also a mere coincidence.

L.146

The authors write that the linear relationship was maintained throughout the evolutionary course for Nc<nc*. but="" fig.2d="" seeing="">Nc* until Delta_X_R reaches the threshold value without taking Delta_X_int into account. Do you think this expectation is correct? If so, the evolutional process is divided into two regimes for Nc>Nc*; in the early stage, the linear relationship holds, and in the late stage after Delta_X_R reaches the threshold value, the linear relationship should be modified to include the effect of Delta_X_int. I think this is natural because in the early stage of evolution the system does not know the existence of the threshold. I do not know if it is the case, but it is desirable to make some comments.

L.174

I think the statement "Thus, the lever principle imposes a fundamental constraint on homeostasis" is too strong. The results presented in this paper are important but still weak for drawing such a conclusion.

Minor

L.22 equivalent -> similar?

L.28 I feel that summary of this last paragraph of Introduction should be added in the abstract, because the results are only very briefly written in the abstract.

L.143 Fig.4C -> 4D?</nc*.>

6. PLOS authors have the option to publish the peer review history of their article (what does this mean?). If published, this will include your full peer review and any attached files.

Reviewer #1: **Yes: **Arnaud Le Rouzic

Reviewer #2: No

---

## [Author Response · Author response to Decision Letter 0]

29 Aug 2022

Below we address the corrections suggested by the reviewers in a point-by-point manner. Note that the reviewers’ comments are noted with brackets, and the corrections are highlighted by red in the revised manuscript.

Reviewer #1:

[ The manuscript is a short report about the evolution of gene expression sensitivity to environmental disturbances in a theoretical gene network model. The authors simulated the evolution of a gene network when a set of genes, named "Core genes", were selected to be expressed at a constant rate whatever the environment. Another set of genes, the "Regulator genes", could evolve in order to help core genes to become stable. The authors shows that core genes were indeed able to evolve robustness, but this evolution was associated with a proportional increase in the sensitivity of regulator genes to the environment. The explanation proposed in the manuscript is that there is a global robustness-plasticity balance in the network, which can be distributed among core and regulatory genes.

As far as I can tell, the authors' results are original, the manuscript is clearly and carefully written (see below for a list of possible misunderstandings), and the conclusions are well-supported. On the down side, the paper is a short report, relatively technical, with little background, and virtually no discussion; it is thus up to the reader to replace the paper in the field, which limits the readership to a relatively small community, and makes it impossible for a non-specialist to assess the interest of the authors' results. As I recently happened to be interested in the very same question in similar theoretical gene network models, I found the results both interesting and insightful, but I am afraid that this opinion might be a bit confidential. 

I have a major concern, and some detailed comments. ]

Thank you for your positive evaluation and constructive suggestions.

[ **** Major concern ****

1) The reference expression x^st(0) for the fitness function (2) conditions the optimal gene expression of the network, but I could not understand how it was determined. Was it computed from an arbitrary genotype at the beginning of the simulation, or was it recomputed for all genotypes? If the first option is true, then the authors need to make it much more explicit. The second option has some deep implications that need to be considered thoroughly. If selection only acts on the robustness of gene expression irrespective to the gene expression level, all non-plastic genotypes are equivalent. However, this will drive the evolution of gene expressions; because of the sigmoid scaling of the gene expression regulation (equation (1)), the sensitivity of the network to the environmental input depends directly on the distance between the sum of J_ij x_j and the threshold. If there is no selection on gene expression, evolution will proceed by maximizing or minimizing (probably minimizing) J_ij x_j to put it as far as possible from the threshold. As a consequence, a network where all core genes were off could be very robust. This could be an explanation for the pattern in fig 3A and 3B, as inhibiting interactions to core genes seem to largely dominate. Yet, I am not sure that a network that would stabilize its expression by shutting down all core genes would be biologically realistic. I assume the authors could track the evolution of average gene expression to ensure that the network is not progressively shutting down to ensure robustness. Constant selection for an arbitrary expression pattern (i.e., drawing randomly the optimal x for each gene at the beginning of the simulation) is also an option. Citing and discussing Siegal & Bergman 2002 (PNAS 99(16), 10528-10532) is probably relevant here. ]

First and foremost, we computed the x^st(0) for each genotype at each generation. This approach might give worry, as you point out, that the initial gene expression itself might have changed so that the expression patterns are less and less sensitive to environmental changes. This is not the case: we set ε and the standard deviation of η much smaller than the threshold (ε = 0.05, σ^2 = 0.01, and y_th = 0.3). When α is zero, i.e., there is no input, all genes are activated only by the noise given by η and spontaneous induction given by ε, so that they fluctuate around x = ε. Then, all genes are normally turned off almost independently of J_ij; since ε / sqrt(N) is 0.005, inputs from 60 genes (60% of the total genes) would have to concentrate on a given gene to exceed the threshold and to turn on a gene constantly without α. Such concentration hardly occurs due to the sparseness of networks (p_link is set to 0.1). Hence, the average gene expression without input did not change through an evolutionary process, and all genes were kept to be off initially. In addition, since the threshold value was fixed, the sensitivity of genes against environmental changes was kept constant. Indeed, in Figs. 2A (before evolution) and 2B (after evolution), gene expression levels fluctuated around ε without input (time = 0 – 100), and those of the core transiently increased in response to changes in α (at time = 100 and 200), indicating that those genes were kept to be sensitive against environmental changes.

This setting might seem biologically unrealistic, but we here focus on the evolution of response against environmental changes and not on the nature of the steady-state gene expression pattern. If we adopt constant selection for an arbitrary expression pattern, it will be difficult to distinguish the effect of the evolution of response from that of the steady state. Thus, we adopt the present simplest settings, in which gene expression pattern is quite stable throughout evolution, and only response can evolve. If we evolve the environmental response from any given stable expression pattern to be robust, inhibiting interactions to core genes will increase so as to suppress a difference of gene expressions from its steady-state level. Then, the pattern in Figs. 3A and 3B will not be due to our setting but will be universal.

We added a detailed explanation of this point in lines 83 and 97.

[ **** Details ****

2) I am not sure that enough details are provided to reproduce the simulations. Some details of the numerical procedure were implicit, and it is probably better if the authors made them explicit. For instance, were numerical simulations repeated several times? They might be affected by a substantial amount of genetic drift, as the population size is rather small (M=300). How was the "steady state expression" obtained? Since the differential equations were stochastic (sigma^2 > 0 in equation (1)), expression never reach a perfect equilibrium (there is always a chance to switch genes on and off randomly), and some threshold was probably used. Such a steady state might not exist for large values of sigma^2, as the network may regularly switch between different equilibria when disturbed. Was reproduction clonal? Was the steady state recomputed in non-mutated offspring? (because of the expression stochasticity, clones might have a slightly different fitness).

As a side note, the authors indicated that "All data are within the manuscript and the Supporting Information files"; I might have missed them but I could not find the corresponding links (to the data and/or the computer code). ]

We repeated the simulation 100 times for each parameter set and plotted the average of those.

Gene expression fluctuated only around the fixed point, as you can see in Figs. 2A and 2B, and switching between on and off hardly occurred due to relatively small noise (σ^2 = 0.01). Then, with a sufficient number of samples, the fluctuation effect should disappear, and thus we can observe the steady state. We computed the steady state by averaging between time = 90 and 100; since we set the time step as 0.01, and then we averaged 1,000 time steps, which will be sufficient to observe the steady-state value.

We set the reproduction as clonal and not sexual, but the mutation was added to J_ij in each generation.

We added a detailed explanation of the above points in lines 124, 83, and 91.

Further, we added the data. 

[ 3) The first part of the results (lines 66 to 84) probably belong to the methods. ]

We reconstructed the manuscript according to your suggestion.

[ 4) Fig 2A and 2B: what does the y axis (labelled "x") stand for? Is it x(i) (gene expression?) or DeltaX (as suggested in the text line 90?). ]

The y axis represents x(i) as the gene expression (x_i(t)). We added explanation to a legend of the figure.

[ 5) I was expecting that the authors discuss the fact that DeltaXc evolves a lot (-100% to -10% depending on Nc) while DeltaXr evolves on a very small scale (0.461 to 0.471, which is about 2%). If plotted on the same scale, the evolution of DeltaXr would be invisible in panel 2D. This asymmetry is not well-represented in fig 4; the drawing features similar weight differences for R genes vs C, while in reality it would be impossible to distinguish the weight difference for regulatory genes. ]

The asymmetry between ΔXc and ΔXr was due to the setting to make gene expression be zero without input and to be one only under sufficiently strong input. Here, since α_1 was set to be larger than the threshold y_th, half of the genes whose h_i had the same sign as α were strongly activated. Then, at the beginning of evolution, when each gene regulatory network was random, only some, but not many, genes were suppressed if inhibitory regulations were concentrated on them. Here, we set α_1 and y_th to 1 and 0.3, respectively, and then only if more than seven inhibitory interactions were concentrated, core genes would be suppressed. Those genes’ expression levels would be intermediate between 0 and 1. Indeed, in Fig. 2A, before evolution, some genes showed intermediate expression level, whereas those of almost all activated genes were around 1. Thus, when the environmental input was sufficiently strong, there was an asymmetry between the numbers of fully activated genes (whose expression level is almost 1) and intermediately activated genes (whose expression level is less than 1). When systems evolved to make robust core genes, inhibition of a large number of fully activated genes mainly contributed to ΔXc. In contrast, activation of the small number of intermediately activated genes mainly contributed to ΔXr. Therefore, the asymmetry between ΔXc and ΔXr emerged.

We added the above explanation in line 163.

[ 6) Line 161, the authors develop an interesting argument: the critical number of core genes is reached when there is not enough R genes to balance the evolution of plasticity. The idea that the ratio Nc/Nr rather than the absolute number of core genes Nc drives the evolution of plasticity is an interesting idea, but this idea could be directly checked by running simulations with a higher number of R genes while maintaining Nc constant. Alternatives include explanations based on genetic load and/or genetic drift (e.g. natural selection in small populations is not strong enough to force robustness on more than Nc genes). If this was true, then Nc should depend on M, not on Nr. ]

To investigate whether the ratio Nc/Nr or the absolute number of core genes Nc determines behaviors of the system, we altered the total number of genes and numerically evolved the system. Then, we found that robustness was scaled by (Nc/N)N^3/4; if only the ratio is the important control parameter, robustness should be scaled by (Nc/N)N^0, or if only the absolute number is important, it should be scaled by (Nc/N)N^1. Then, the actual scaling was intermediate between these two cases. This suggests that the balance between core and regulatory genes depends not simply on the number of genes.

We added the results (Fig. 5) and noted this point in line 262.

 

Reviewer #2:

[ Summary:

The authors conducted evolutional simulations for a model of gene regulatory networks, which consisted of the core genes and the regulatory genes. As a fitness, they required the homeostasis of the core genes. For the evolved networks, they found that the robustness (homeostasis) of the core genes and plasticity of the regulatory genes were reciprocally correlated. Thus, the robustness of the core genes is compensated by the plasticity of the regulatory genes. In case the ratio of core genes is higher than the threshold value, this compensation was incomplete. They also observed the lever principle held between these two.

I think these results are novel and interesting enough, and will induce future research both in computational and experimental biology. Thus, it is worth publishing in PLOS ONE, provided the following points are made clear. ]

Thank you for your positive evaluation and constructive comments.

[ Major

L.96

The authors write that there is a transition from the perfect adaptation to the partial adaptation, and the critical value of Nc lies between 20 and 25. However, I do not observe the transition in Fig.2C; the lines for Nc/N=0.2 to 0.6 do not reach steady-state within 1000 generations and thus it is difficult to distinguish whether the steady-state value gradually changes or there indeed is the transition. This point should be clarified, for example by presenting the steady-state values for Nc/N=0.2 to 0.6. Whether this transition point agrees with Nc/N that Delta_X_R reaches the threshold value before Delta_X_C reaches zero should be clarified (although written briefly in L.106). ]

Thank you for raising an important point that the relaxation was slowed down for Nc/N=0.2 to 0.6, in particular between Nc/N=0.2 and 0.3. In statistical physics, it is commonly observed that the relaxation of systems to the equilibrium is slowed down in the vicinity of the transition point. We considered that slowing down, you pointed out, is the critical slowing down and can be evidence of a transition between perfect and partial adaptation regimes. Since it will take a very long time to relax to the steady state in the vicinity of the transition point, it will be difficult to clarify the actual transition point numerically. However, it is clear from Fig. 2D that ΔX^C will reach zero in the extension of the evolutionary trajectory for Nc/N=0.2, whereas ΔX^C will never reach zero for Nc/N=0.3. Thus, it would be clear that the transition point lies between Nc/N=0.2 and 0.3, even though the relaxation to the steady state was not complete. 

We clearly describe this point in the revised text in line 133.

[ L.106

The authors write that there is a threshold value near 0.471 and I notice there indeed is the threshold. Intuitively, this threshold value is a consequence of the stochasticity of evolution and depends on beta_evo. Do you think this intuition correct, or there is other reason for this threshold? The author should mention the meaning or origin of this threshold value at least qualitatively. ]

We consider that the threshold value will depend little on beta_evo and strongly on the detail of the dynamics. Here, the threshold value depends on the stochastic dynamics of gene expression; if there is no fluctuation, the steady state value of each x is ε without environmental input, and 1 + ε under the input α with the same sign to h_i. Because half of the genes have positive h_i, and another half have a negative one, the threshold value will be estimated as 0.5 (= (1 + ε - ε)^2 / 2.0). However, because of fluctuation and a constraint to keep gene expression positive, the steady state value of x without input would be higher than ε; x reached about 0.3 by fluctuation but was truncated at 0 at the bottom, as shown in Figs. 2A and 2B, while we set ε as 0.05. Then, the noise in gene expression was asymmetrical in the vicinity of x = 0. Thus, the threshold value cannot reach 0.5 and will be slightly lower than 0.5.

We added an explanation of this point in line 149.

[ L.127

In Fig.3E, the value starts to decrease near Nc=30, and in Fig.3E, peaks are also near Nc=30. Is this a mere coincidence or do they have some common reason? Also, Nc=30 is close to the transition point mentioned before. Is this also a mere coincidence. ]

As mentioned above, the transition point from perfect to partial adaptation was in the vicinity of Nc=30, and the characteristics you pointed out will be deeply related to the transition. In the perfect adaptation region, the core genes were perfectly inhibited by regulatory genes, and if the regulation from the regulatory genes was lacking, the core genes never showed adaptation (below Nc = 30 in Fig. 3E). At the same time, if Nc increased, the contribution of each regulatory gene to repress core genes increased, and then flipping the expression of one regulatory gene had a great influence (Fig. 3F). In contrast, in the partial adaptation region, the core genes regulated each other because the number of regulatory genes was insufficient. Then, these showed partial robustness without the regulation from the regulatory genes (above Nc = 30 in Fig. 3E). Moreover, if Nc increased in the partial adaptation region, the mutual regulation within the core genes became relatively stronger than the regulation from the regulatory genes, then the influence of a flip of one regulatory gene decreased. Thus, we expect that the influence of flipping the expression of one regulatory gene will show the peak around the transition point from perfect to partial adaptation. 

We added an explanation on this point in lines 189 and 204.

[ L.146

The authors write that the linear relationship was maintained throughout the evolutionary course for NcNc* until Delta_X_R reaches the threshold value without taking Delta_X_int into account. Do you think this expectation is correct? If so, the evolutional process is divided into two regimes for Nc>Nc*; in the early stage, the linear relationship holds, and in the late stage after Delta_X_R reaches the threshold value, the linear relationship should be modified to include the effect of Delta_X_int. I think this is natural because in the early stage of evolution the system does not know the existence of the threshold. I do not know if it is the case, but it is desirable to make some comments. ]

For Nc < Nc*, as you expected, evolution of the regulation from the regulatory to core genes was sufficiently fast, and the linear relationship was observed until Delta_X_R reaches the threshold value. However, for Nc > Nc*, evolution of intra-connections among the core genes was also fast and it was difficult to observe the clear linear relationship. If the evolution of the regulations from the regulatory genes were much faster than the intra-connections among the core genes, the linear relationship would hold in the early stage, as you described. However, the timescale of evolution depends on Nc; if Nc is much larger than Nr, the number of intra-connections among the core genes is much larger than that of interconnections between core and regulatory genes. Here, the intra-connections can change with higher probability by the mutation in each generation. Then, intra-connections will typically change faster than the interconnections and evolve faster. Then, we cannot observe the clear linear relationship for Nc > Nc*.

We added a comment on this point in the footnote.

[ L.174

I think the statement "Thus, the lever principle imposes a fundamental constraint on homeostasis" is too strong. The results presented in this paper are important but still weak for drawing such a conclusion. ]

We toned down the conclusion.

[ Minor

L.22 equivalent -> similar? ]

We revised the expression according to your suggestion.

[ L.28 I feel that summary of this last paragraph of Introduction should be added in the abstract, because the results are only very briefly written in the abstract. ]

We added some sentences in the indicated paragraph to the abstract.

[ L.143 Fig.4C -> 4D? ]

We corrected the typo.

---

## [Decision Letter · Decision Letter 1]

12 Sep 2022

PONE-D-22-14450R1A Linear Reciprocal Relationship Between Robustness and Plasticity in Homeostatic Biological NetworksPLOS ONE

Dear Dr. Hatakeyama,

Thank you for submitting your manuscript to PLOS ONE. After careful consideration, we feel that it has merit but does not fully meet PLOS ONE’s publication criteria as it currently stands. Therefore, we invite you to submit a revised version of the manuscript that addresses the points raised during the review process.

We look forward to receiving your revised manuscript.

Kind regards,

Claus Kadelka

Academic Editor

PLOS ONE

Journal Requirements:

Additional Editor Comments:

Thank you for your efforts resulting in this improved manuscript. Please ensure all your code and data is made publicly available for reproducibility as outlined in the PLOS Data Policy. Also, please address the second comment by the reviewer prior to resubmission.

Reviewers' comments:

Reviewer's Responses to Questions

**Comments to the Author**

1. If the authors have adequately addressed your comments raised in a previous round of review and you feel that this manuscript is now acceptable for publication, you may indicate that here to bypass the “Comments to the Author” section, enter your conflict of interest statement in the “Confidential to Editor” section, and submit your "Accept" recommendation.

Reviewer #1: (No Response)

Reviewer #2: All comments have been addressed

2. Is the manuscript technically sound, and do the data support the conclusions?

Reviewer #1: Yes

Reviewer #2: (No Response)

3. Has the statistical analysis been performed appropriately and rigorously? 

Reviewer #1: N/A

Reviewer #2: (No Response)

4. Have the authors made all data underlying the findings in their manuscript fully available?

Reviewer #1: No

Reviewer #2: (No Response)

5. Is the manuscript presented in an intelligible fashion and written in standard English?

Reviewer #1: Yes

Reviewer #2: (No Response)

6. Review Comments to the Author

Reviewer #1: Review of "A linear reciprocal relationship between robustness and plasticity in homeostatic biological networks" by T.S. Hatakeyama and K. Kaneko, PONE-D-22-14450R1.

This is the revision of a manuscript I reviewed a few months ago (PONE-D-22-14450). The scientific content of the manuscript was sound, and in spite of a rather technical content, I found it suitable for publication in PLoS ONE. I had a major concern and a few comments, each having being addressed in the response to the reviewers, and in the text of the paper. The most significant changes were in the abstract (which as mostly rewritten), in the methods (many of the reviewers' comments were related to the methods), and in the results/discussion, with additional explanations provided in 6 paragraphs, as well as an additional figure (new figure 5). This revision is substantially longer than the original version (~ 2 pages), and has the same pros (scientifically sound) and cons (probably accessible to specialists only) than the first submission. Most of the changes in the manuscript are verbatim copies of the responses to the reviewers, which is not problematic per se (although it does not help to make the manuscript more accessible). Note that data/software sharing is minimal, and (in my opinion) does not match at all PLOS transparency guidelines.

About my comment #1 (what the optimal gene expression fixed at the beginning of the simulations), the authors now make it clear how the optimal expression was determined. Although I may not necessarily agree with the modeling choice (which implies that the network is completely shut down when the environmental signal is at zero), the authors now discuss the consequences of this model setting, and cite the paper by Siegal & Bergman about the relationship between stabilizing selection for an optimum and selection for network stability, which is probably more than enough to inform the reader about potential pitfalls.

About my comment #2 (not enough details to reproduce the simulations): the authors have expanded the model description, and it is probably easier now to re-implement the model. Note that the author's statement about data availability states that "data are fully available without restriction", which I find misleading: the authors have provided only summary statistics (basically, the location of the points on the figures). As far as I can tell, there is nothing available about the simulation code (the implementation of the model), nothing about the analysis code (how data was extracted from the simulation output, how summary statistics were computed from the 100 replicates, etc). This is as far to "open science" as possible, and, as far as I can tell, contradicts directly PLoS One policies about data sharing (https://journals.plos.org/plosone/s/data-availability) and software sharing (https://journals.plos.org/plosone/s/materials-software-and-code-sharing).

My comment #3 (structure of the manuscript) has been fully addressed.

My comment #4 has been addressed (in the caption of the figure, not in the figure itself, not a big deal).

My comment #5 (range of evolution for DeltaX) is now addressed in the manuscript.

My comment #6 has been addressed in a very satisfactory way, with a new figure and more insightful results.

In the text: line 91, "We set the asexual reproduction" sounds awkward (perhaps "Reproduction was asexual"?).

Signed: Arnaud Le Rouzic

Reviewer #2: (No Response)

7. PLOS authors have the option to publish the peer review history of their article (what does this mean?). If published, this will include your full peer review and any attached files.

Reviewer #1: **Yes: **Arnaud Le Rouzic

Reviewer #2: No

---

## [Author Response · Author response to Decision Letter 1]

20 Oct 2022

Below we address the corrections suggested by the reviewers in a point-by-point manner. Note that the reviewers’ comments are noted with brackets, and the corrections are highlighted by red in the revised manuscript.

Reviewer #1:

[ This is the revision of a manuscript I reviewed a few months ago (PONE-D-22-14450). The scientific content of the manuscript was sound, and in spite of a rather technical content, I found it suitable for publication in PLoS ONE. I had a major concern and a few comments, each having being addressed in the response to the reviewers, and in the text of the paper. The most significant changes were in the abstract (which as mostly rewritten), in the methods (many of the reviewers' comments were related to the methods), and in the results/discussion, with additional explanations provided in 6 paragraphs, as well as an additional figure (new figure 5). This revision is substantially longer than the original version (~ 2 pages), and has the same pros (scientifically sound) and cons (probably accessible to specialists only) than the first submission. Most of the changes in the manuscript are verbatim copies of the responses to the reviewers, which is not problematic per se (although it does not help to make the manuscript more accessible). Note that data/software sharing is minimal, and (in my opinion) does not match at all PLOS transparency guidelines. ]

Thank you for your comments.

As you pointed out, it is harder for non-specialist to access our revised manuscript than the first submitted version. Thus, we moved some sentences to the reference list (as a footnote) to improve the readability.

[ About my comment #1 (what the optimal gene expression fixed at the beginning of the simulations), the authors now make it clear how the optimal expression was determined. Although I may not necessarily agree with the modeling choice (which implies that the network is completely shut down when the environmental signal is at zero), the authors now discuss the consequences of this model setting, and cite the paper by Siegal & Bergman about the relationship between stabilizing selection for an optimum and selection for network stability, which is probably more than enough to inform the reader about potential pitfalls. ]

Thank you for your positive evaluation.

[ About my comment #2 (not enough details to reproduce the simulations): the authors have expanded the model description, and it is probably easier now to re-implement the model. Note that the author's statement about data availability states that "data are fully available without restriction", which I find misleading: the authors have provided only summary statistics (basically, the location of the points on the figures). As far as I can tell, there is nothing available about the simulation code (the implementation of the model), nothing about the analysis code (how data was extracted from the simulation output, how summary statistics were computed from the 100 replicates, etc). This is as far to "open science" as possible, and, as far as I can tell, contradicts directly PLoS One policies about data sharing (https://journals.plos.org/plosone/s/data-availability) and software sharing (https://journals.plos.org/plosone/s/materials-software-and-code-sharing). ]

Since the total number of files is huge (100 files for each point in each figure), we uploaded the averaged data as samples in the last time. In the revised manuscript, we uploaded the codes for Fig. 2AB, for Figs. 2CD, 4, and 5, for Fig. 3ABCD, for Fig. 3E, and for Fig, 3F, to assure the reproducibility.

[ My comment #3 (structure of the manuscript) has been fully addressed.

My comment #4 has been addressed (in the caption of the figure, not in the figure itself, not a big deal).

My comment #5 (range of evolution for DeltaX) is now addressed in the manuscript.

My comment #6 has been addressed in a very satisfactory way, with a new figure and more insightful results. ]

Thank you for your positive comments.

[ In the text: line 91, "We set the asexual reproduction" sounds awkward (perhaps "Reproduction was asexual"?). ]

We have corrected the sentence.

---

## [Editor Report · Decision Letter 2]

24 Oct 2022

A Linear Reciprocal Relationship Between Robustness and Plasticity in Homeostatic Biological Networks

PONE-D-22-14450R2

Dear Dr. Hatakeyama,

We’re pleased to inform you that your manuscript has been judged scientifically suitable for publication and will be formally accepted for publication once it meets all outstanding technical requirements.

Kind regards,

Claus Kadelka

Academic Editor

PLOS ONE

Additional Editor Comments (optional):

Thank you for attaching data and code files. This ensures, as required by PLOS, that your results one be reproduced.